# Short-Term Electrical Load Forecasting Based on VMD and GRU-TCN Hybrid Network

**Changchun Cai** [1,2,*,†], **Yuanjia Li** [1,2], **Zhenghua Su** [3], **Tianqi Zhu** [1,2] and **Yaoyao He** [1,2]

1 Jiangsu Key Laboratory of Power Transmission and Distribution Equipment Technology, Hohai University, Changzhou 213022, China; 18739086714@163.com (Y.L.); zhutq@163.com (T.Z.); heyy@hhu.edu.cn (Y.H.)
2 College of The Internet of Things Engineering, Hohai University, Changzhou 213022, China
3 State Grid Changzhou Power Supply Company, Changzhou 210024, China; gerrysu@163.com
* Correspondence: 20031690@hhu.edu.cn
† Member, IEEE.

**Abstract:** With the continuous increase in user-side flexible controllable resources connected into a distribution system, the components of electrical load become too diverse and difficult to be accuracy forecasted. A short-term load forecast method that integrates variational modal decomposition (VMD), gated recurrent unit (GRU) and time convolutional network (TCN) into a hybrid network is proposed in this paper. Firstly, original electrical load sequence data with noise are decomposed into intrinsic IMF components with different frequencies and amplitudes based on the VMD method. Secondly, a combined load forecasting method based on the GRU and TCN network is proposed for the high and low-frequency load subsequent signals, respectively. Finally, the high and low-frequency signals forecasting results of the GRU and TCN network are rebuilt for the final load forecasting. The experiment results based on actual operation data (data set 1) and simulation data (data set 2), which show that the proposed method can reduce the forecasting error by 36.20% and 10.8%, respectively, in comparison with VMD-GRU. The reliability and accuracy of the proposed method is verified through the comparison with other methods such as LSTM, Prophet and XG Boost.

**Keywords:** short-term load forecasting; variational modal decomposition (VMD); gated recurrent unit (GRU); time convolutional network (TCN); hybrid algorithm

## 1. Introduction

With the presentation of the goal of "carbon peaking and carbon neutrality" in China, the construction of a new electrical power system based on new energy is imminent. Under the new electrical power system architecture, the interaction of renewable energy generations, energy storage and flexible load resources is becoming more frequent. The accuracy and reliable challenges posed by the higher penetration of renewable energy and flexible load resources should be considered significantly. Short-term load forecast is the basis work of power grid dispatching, and it is becoming more important in the new electrical power system [1–3].

Various short-term load forecasting models and methods have been proposed and applied during the past decades. Typically, load forecasting methods are divided into statistical-based and machine learning-based forecasting methods [4–7]. Statistical methods include multiple linear regression [8], autoregressive moving average [9], and autoregression [10,11]. These methods have a distinctive feature with faster calculation speed. However, they also have a weak expressive ability for nonlinear problems. With the development of artificial intelligence technology recently, machine learning methods such as gray system [12], recurrent neural network [13], and extreme learning machine [14] are focused and widely used in load forecasting areas. However, there are gradient extinction and gradient explosion problems in the back propagation of recurrent neural network with the layer increase in the deep structure. Extreme learning machine is a combined stochastic

prediction and multiple regression method; its performance reduces with the increasing complexity of load forecasting considering various meteorological and electrical consumption behaviors. The k-fold cross-validation estimator is widely used in classification and regression problems. A global support vector regression metamodeling approach based on the sorted k-fold method is proposed in [15], in which the k number of SVR metamodels are constructed based on a cross-validation technique. A k-fold cross-validation approach based on the DCNN algorithm is used for predicting the melanoma classes in [16], and the k-fold method can achieve a better performance on a small skin cancer data set.

Furthermore, load forecasting methods based on combined algorithms emerge continuously, which have achieved results with a faster calculation speed and higher precision. A deep learning bidirectional LSTM neural network is proposed in [17]; the method gives a deep learning structure of bidirectional LSTM to avoid the gradient explosion problem. A hybrid short-term load forecasting is proposed in [18], which uses the combination of correlation analysis and appropriate inputs to the individual Bayesian neural network, but the parallel training and its weight of Bayesian neural network introduce an extra computational burden. A novel hybrid with a new signal decomposition and correlation analysis technique method is proposed in [19]; T-copula is used to compensate the peak load to reduce the forecasting error. A combined method based on the maximum information coefficient, factor analysis, gray wolf optimization and generalized regression neural network is proposed in [20]; the proposed method has a strong global search ability, but it needs enough time for the training and forecasting.

The variation of electrical price will influence the consumption of electrical power, which depends on the electrical power curves. So, the electrical price is a key point for the load forecasting. A new approach combining parametric and nonparametric function autoregressive models for the prediction of electricity price is proposed in [21]; it provides insights into the scale and purchase mechanism with the price and demand formation process. The paper provides a new solution from the purchase curve side to improve the forecasting accuracy. A load forecasting model based on least square support vector machine and chaotic time series is proposed in [22], and the real-time price of power load and chaotic characteristics is considered. A short-term load forecasting method based on price-responsive load is proposed in [23], and the response of the aggregate load to price is modeled by a set of marginal utility curves and power consumption limitations. The price response factors are estimated using a generalized inverse optimization scheme.

Power load time-series data is a non-stationary random process, which has the characteristics of quasi-periodicity, non-stationarity and nonlinearity. In order to improve the stationarization of power load time-series data, empirical mode decomposition (EMD) [24] and the frequency domain decomposition algorithm [25] are used to decompose the original load time series. A data-driven deep learning framework to load forecasting is proposed in [26], Box–Cox transformation is used to process data, and deep belief network is used for the load forecasting. A user-side load prediction method that integrates empirical mode decomposition and deep belief network is proposed in [27]; by decomposing the original load data into several eigenmode functions with different frequencies and amplitudes, the deep belief network is used to feature extraction and time-series prediction for each modal function. A load forecasting method based on ensemble empirical mode decomposition (EMMD) and Elman network is proposed in [28], the EEMD sample entropy was used to decompose the original power load sequence into a series of subsequences. A fault feature extraction method based on VMD optimized with information entropy is proposed in [29], and several intrinsic model functions were obtained by signal decomposition. Variable mode decomposition is used to generate 2D time frequency spectrograms from the various fault conditions of bearing in [30], and deep convolutional neural network is applied for fault diagnosis.

With the development of deep learning theory in the field of time-series data processing, CNNs have an outstanding performance in comparison with traditional RNN models such as LSTM. It is believed that sequence tasks should be reconsidered with RNN and

the CNN method. The CNN neural network is seen as a new solution for the sequence modeling tasks [31]. Furthermore, a temporal convolutional network (TCN) is proposed, which accelerates the feedback and convergence of the deep network through the setting of the residual unit, and it solves the degradation phenomenon caused by the increase in the network level [32]. A short-term load forecasting model for industrial customers based on the temporal convolutional network (TCN) and Light Gradient Boosting Machine (LightGBM) is proposed in [33]; the proposed method can extract the hidden information and long-term temporal relationships in the electrical load. In [34], convolution neural networks (CNNs) and gated recurrent unit (GRU) is combined for probabilistic residential load forecasting, and a deep model based on the mixture density network is used to directly predict probability density functions. An improved stacked gated recurrent unit-recurrent neural network (GRU-RNN) for both univariate and multivariate scenarios is proposed in [35]; the modified algorithm establishes an accurate mapping between the selected variables and load through its self-feedback.

In this paper, a hybrid load forecasting method based on variational modal decomposition (VMD), temporal convolutional network (TCN) and gated recurrent unit (GRU) is proposed, and the contributions of this paper are as follows: (1) Variation modal decomposition is used for the original electrical load decomposition with different frequencies. (2) A combined load forecasting method based on TCN and GRU neural network is proposed in this paper, the two neural networks are used for the forecasting of different IMF components based on its frequency characteristics, and it can extract the characteristics of electrical load in different frequency modes. (3) The simulation based on the actual electrical load is implemented in this paper; the proposed method can improve the forecast result both in accuracy and speed.

The structure of the paper is arranged as follows: The variational modal decomposition method and its application in the power load are discussed in Section 2. In Section 3, the hybrid neural network method based on GRU and TCN is introduced, and the reasons for the application of each neural network for the low and high-frequency load decomposition signal are discussed. In Section 4, two experiment data sets are used to verify the proposed method; the results show that the proposed method can catch the dynamic of electrical load in different frequency bandwidths and improve the accuracy of the electrical load forecasting.

## 2. Variational Modal Decomposition of Electrical Load

With the deepening of the re-electrification process of the energy system, more and more renewable generations will connect into power system. The power consumption characteristics in the user side are changing significantly. It is difficult to forecast the load profile in the power station with large-scale renewable generations. On the one hand, the output power of renewable energy is volatile, which causes fluctuation of the power load in the power station. On the other hand, the power load data contain a lot of random noise from users' individual electricity habits. For example, the yearly and daily periodicities can be seen in Figure 1.

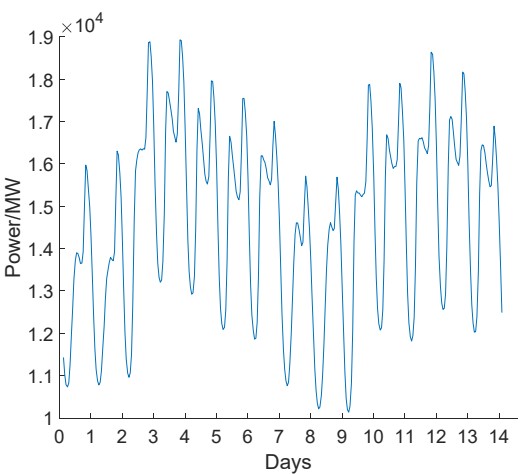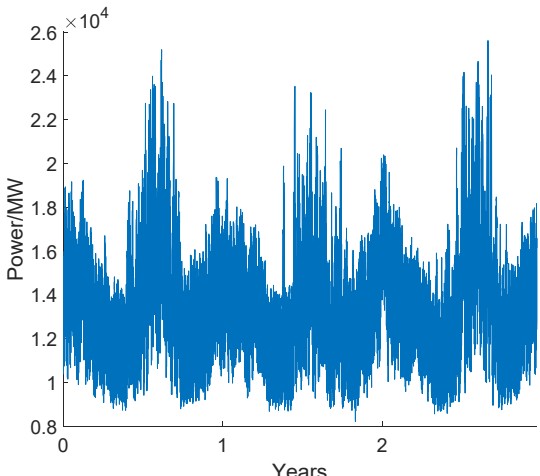

**Figure 1.** Yearly seasonality load profiles.

Load profile is essentially a comprehensive reflection of the dynamic response of various electrical resources in the load-side system. The principle output power of different components is similar and can be distinguished easily. It is mature to forecast a single specific component based on its physical mechanism. Therefore, the load forecast accuracy can be improved by decomposing the total load profile into a series of component levels with certain characteristics. VMD is an adaptive, non-recursive method for modal variation and signal processing. The number of modal decompositions of the signal sequence is determined by the change of the actual signal sequence. The optimal frequency center and limited bandwidth of each mode are adaptive matched and iterated to separate the intrinsic modal components (IMFs) and residuals. As a result, the frequency domain division of the power load time-series signal is obtained.

It assumes that all signal sub-components are narrowband signals which concentrated around their respective center frequencies in the VMD method. As a result, a constrained optimization problem based on the components' narrowband condition is established. The center frequencies of the different components and reconstructing their respective components is the core of VMD. It can effectively reduce the complexity of decomposition for the non-stationary random sequences compared with the EMD, and the subsequences have different frequency scales and stationary characteristics. The establishment and optimization of the Variational Problem (VP) are the two cores of the VMD method.

*2.1. Variational Problem Establishment*

In this paper, the alternating direction multiplier method is used to continuously update each mode and its center frequency. The corresponding fundamental frequency band of each mode is gradually demodulated to obtain the center frequency. We assume that the original signal sequence is $f(t)$, $K$ is the number of center frequencies, and $\{u_k(t)\}$ represent the components of IMF in the limited bandwidth. Based on the Hilbert transform, the one-sided spectral signal of envelope signal $u_k(t)$ can be rewritten as $\left[\delta(t) + \frac{j}{\pi t}\right] * u_k(t)$. The frequency of each IMF is modeled to the baseband based on the center frequency $\omega_k$. It can be written as $\left[\left(\left[\delta(t) + \frac{j}{\pi t}\right]\right) * u_k(t)\right] e^{-j\omega_k t}$.

The Variational Problem (VP) based on each IMF component signal bandwidth can be written as

$$\min_{\{v_k\},\{\omega_k\}} \left\{ \sum_{k=1}^{K} \|\partial_t \left[ \left( \delta(t) + \frac{j}{\pi t} \otimes v_k(t) \right) \right] e^{-j\omega_k t} \|_2^2 \right\} \tag{1}$$

$$\text{s.t.} \sum_{k=1}^{K} u_k = f \tag{2}$$

where $\{u_k\}$ is the IMF component, $\{\omega_k\}$ is the center frequency of each IMF component, $\otimes$ is the convolution operator, $\delta(t)$ is the Dirac distribution, and $\partial_t$ is the partial derivative function.

Lagrange Multiplier $\lambda(t)$ and bandwidth parameters $\alpha$ are introduced to turn the constrained Variational Problem into an unconstrained Variational Problem, and the extended Lagrangian function for the Variational Problem can be written as:

$$L(\{u_k\},\{\omega_k\},\lambda) = \alpha\sum_k \left\| \partial_t \left[ \left( \delta(t) + \frac{j}{\pi t} \right) * u_k(t) \right] e^{-j\omega_k t} \right\|^2 \\ + \left\| f(t) - \sum_k u_k(t) \right\|_2^2 + \; < \lambda(t), f(t) - \sum_k u_k(t) > \tag{3}$$

### 2.2. Variational Problem Solution

The IMF component $u_k(t)$ and center frequency $\omega_k(t)$ of each mode are updated based on the alternating direction multiplier method, and the new modal components can be written as:

$$\hat{u}_k^{n+1}(\omega) = \frac{\hat{f}(\omega) - \sum\limits_{i\neq k} \hat{u}_i(\omega) + (\hat{\lambda}(\omega)/2)}{1 + 2\alpha(\omega - \omega_k)^2} \tag{4}$$

where, $\hat{f}(\omega)$, $\hat{u}_i(\omega)$, $\hat{\lambda}(\omega)$ and $\hat{u}_k^{n+1}$ are the estimated value of $f(t)$, $u_t(t)$, $\lambda(t)$ and $u_k^{n+1}$ after Fourier transform, $n$ is the number of iterations, and $\varpi$ is the frequency. The VMD calculation process is shown in Figure 2, and parameter estimation can be written as:

$$\hat{\omega}_k^{n+1} = \frac{\int_0^\infty \omega |\hat{u}_k(\omega)|^2 d\omega}{\int_0^\infty |\hat{u}_k(\omega)|^2 d\omega} \tag{5}$$

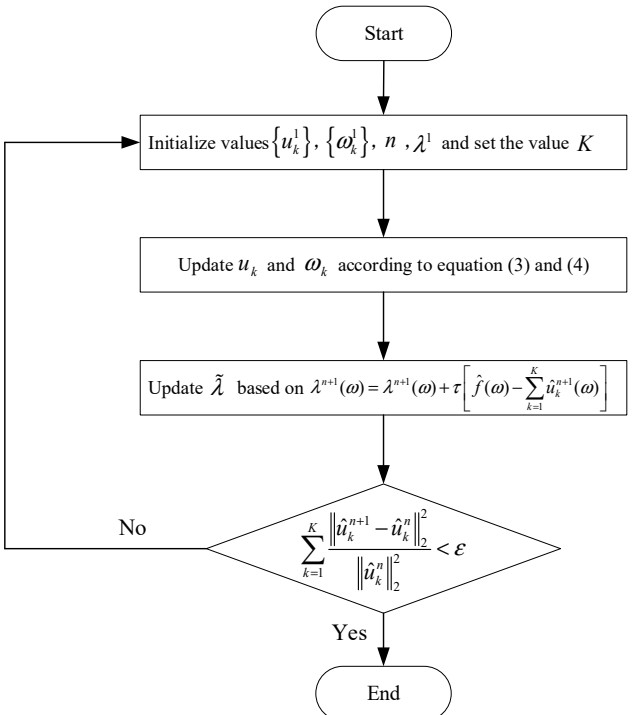

**Figure 2.** VMD calculation process.

## 3. Load Forecasting Based on GRU and TCN Hybrid Network

### 3.1. Gate Recurrence Unit (GRU)

A long short-term memory (LSTM) network can solve the gradient explosion and gradient disappearance problem in RNN, and LSTM has been widely used in the area of data processing in recent years. GRU is a variant of LSTM with fewer internal units; it

combines the input gate and forget gate into one update gate. It is easier to perform the training process and save computing time compared with LSTM. The substructure of GRU is shown in Figure 3; there are only two gates: namely, the update gate and the reset gate. Its mathematical description is shown in Formula (6).

$$
\begin{cases}
r_t = \sigma(W_r \cdot [h_{t-1}, x_t]) \\
z_t = \sigma(W_z \cdot [h_{t-1}, x_t]) \\
\widetilde{h}_t = \phi(W_{\overline{h}} \cdot [r_t \times h_{t-1}, x_t]) \\
h_t = (I - z_t) \times h_{t-1} + z_t \times \widetilde{h}_t \\
y_t = \sigma(W_o \cdot h_t)
\end{cases}
\tag{6}
$$

where $z_t$ and $r_t$ are the update gate and reset gate, $\sigma$ is the Sigmoid function, $\phi$ is the hyperbolic tangent function, $x_t$, $h_{t-1}$, $h_t$ and $y_t$ are the vector of input data, state memory variable in time $t-1$ and $t$, and output data, respectively. $W_r$, $W_z$, $W_{\overline{h}}$ and $W_o$ are the weight factor of the connection matrix between $x_t$ and $h_{t-1}$ in the update gate, reset gate, memory unit, and output gate, respectively. $I$ is the identity matrix. The update gate is used to control the information percentage of the previous moment when it is brought into the next moment. The reset gate controls how much information from the previous state is reserved to the current candidate set $\widetilde{h}_t$.

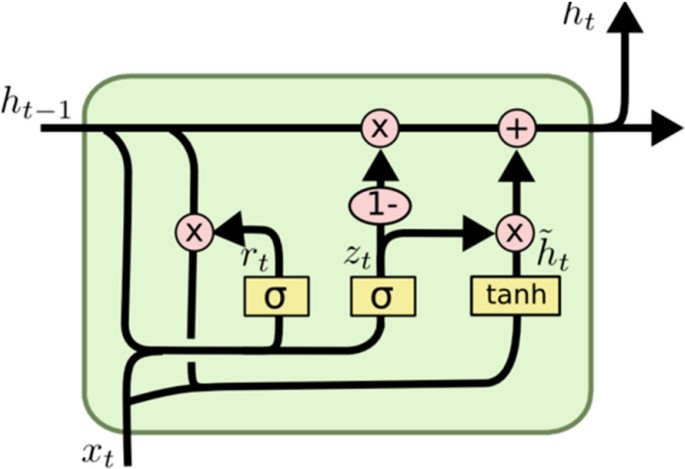

**Figure 3.** Basic structure of GRU.

### 3.2. Temporal Convolutional Network

Temporal convolutional network (TCN) is a variant of convolutional neural network (CNN). The core of TCN is atrous convolution and residual block. TCN has a larger causal convolution than traditional causal convolution with the same network depth, and it can flexibly adjust the size of the receptive field. The TCN model solves the gradient explosion and gradient disappearance problem by initializing weight parameters and regularization layers in time-series data prediction. The residual module is used to make the network information directly and deeply transmitted through the cross-layer form in TCN.

The convolution process of standard convolution and atrous convolution is shown in Figure 4. By injecting holes into the convolution, atrous convolution allows interval sampling in the input data, which increases the receptive field without going through the pooling layer. The convolution output contains a large range of information to ensure the authenticity of the information.

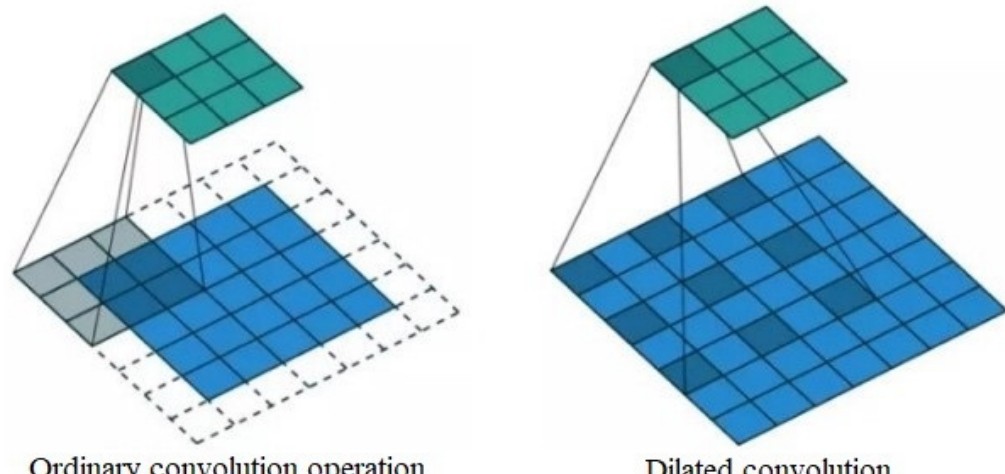

**Figure 4.** Comparison of CNN and TCN convolution.

The receptive field can be written as:

$$R = k_s d_{\max} s \tag{7}$$

For 1D sequence input $X$ and filter $f$, atrous convolution operation $F$ on the input sequence can be written as:

$$F(s) = \sum_{i=0}^{k-1} f(i) X_{s-d \cdot i} \tag{8}$$

where $R$ is the size of the receptive field, $k_s$ is the size of the convolution kernel, $d_{\max}$ is the maximum number of hollows, $s$ is the number of convolutional layers, $d$ is the hollow coefficient, $k$ is the filter size, and $s - d \cdot i$ is the direction of past data. It can be seen that the receptive field is positively correlated with the size of the convolution kernel, the maximum number of holes and the number of convolution layers.

The residual connection module uses the previous information and converted them as the current input to ensure the training efficiency; the residual connection module can be written as:

$$o = Activation(x + \chi(x)) \tag{9}$$

where $o$ is the output of residual block, *Activation* is the activation function, $x$ is previous information, and $\chi(x)$ is converted information.

### 3.3. Load Forecasting Based on VMD-GRU-TCN

After VMD decomposition, different IMF components have different frequency characteristics. In this paper, different neural networks are selected for the data training and load forecasting based on the different frequency characteristics of the load modal components. As mentioned above, the GRU network can better capture the time-series characteristics and variation rules of stationary signals, and it can obtain higher prediction accuracy when the model parameters do not change much. It is suitable for the forecasting of the low-frequency load components. The convolutional layer of TCN has both atrous convolution and causal convolution structures. Atrous convolution allows the volume to obtain different receptive fields by controlling the sampling rate; it covers a suitable range of sequence values. In addition, through the parallel performance of hole convolution, the time-series tracking of high-frequency fluctuation signals can be improved significantly. The forecasting results of the high and low-frequency load components are rebuilt for the result of the load forecast; the process of the proposed method is shown in Figure 5. The specific forecast steps are as follows:

(1)   Data preprocessing. The original load data contain abnormal data and missing data. First, the data are cleaned by DBSCAN clustering, the abnormal data are eliminated and the missing data are supplemented.

(2)   Decomposition of time-series load data. The load time series data are decomposed by VMD to obtain the different $f(t)$, $u_k(t)$, and $\lambda(t)$, corresponding to the characteristic data with different frequencies.

(3)   Load forecasting. We construct a load forecasting model based on the GRU network and TCN network, select the load data on similar days to determine the training date and carry out model training to obtain the load forecasting model; then, the load data of different components in the future day are forecasted.

(4)   Load reconstruction. The forecast data of different components are superimposed and reconstructed to obtain the forecast daily load data.

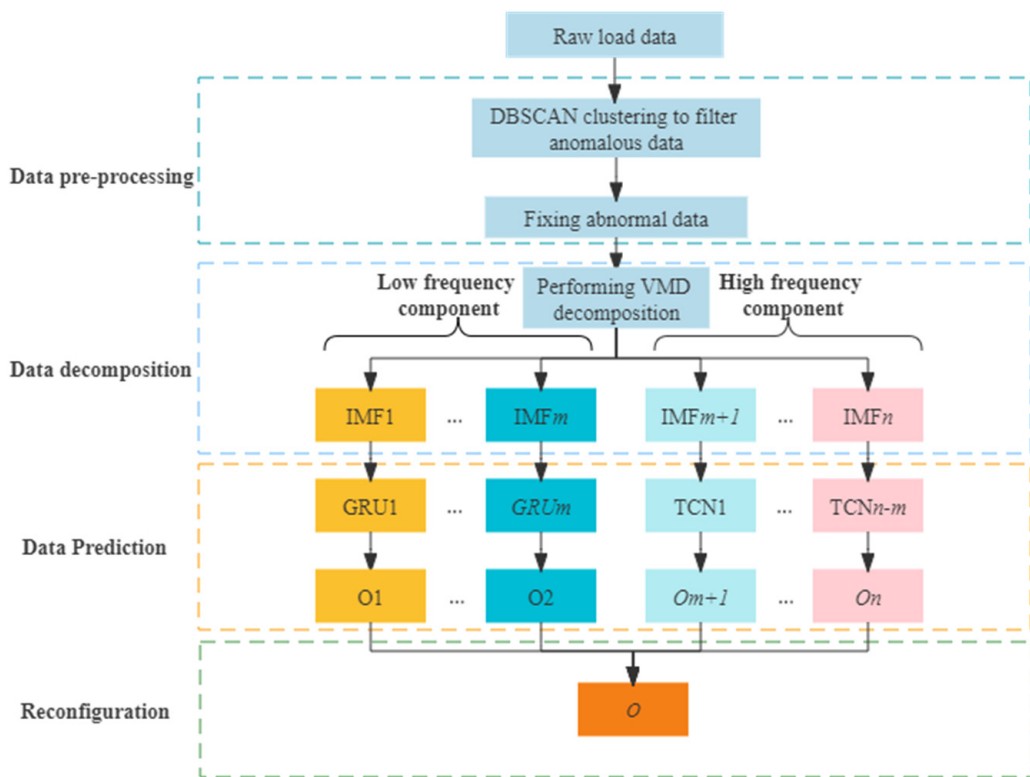

**Figure 5.** VMD-GRU-TCN based forecasting process.

## 4. Experiment Simulation and Data Analysis

### 4.1. Experimental Environment and Load Data

The experiment was completed using Python 3.8 n and the open source library vmdpy to write the VMD program. The neural network was implemented in Keras 2.6. The data set contains two parts in this experiment: the GEFCom2012 [36] load data and the actual load data of the power station in the southwest province in China during 1 January 2016 to 31 December 2018. The sampling time of the data set is 15 min.

Firstly, the load data are classified by the density-based clustering algorithm (DBSCAN). The clustering result is shown in Figure 6. Normal data are divided into one cluster, and the remaining clusters can be regarded as abnormal data. For anomalous data, it will be replaced by the mean of before and after data.

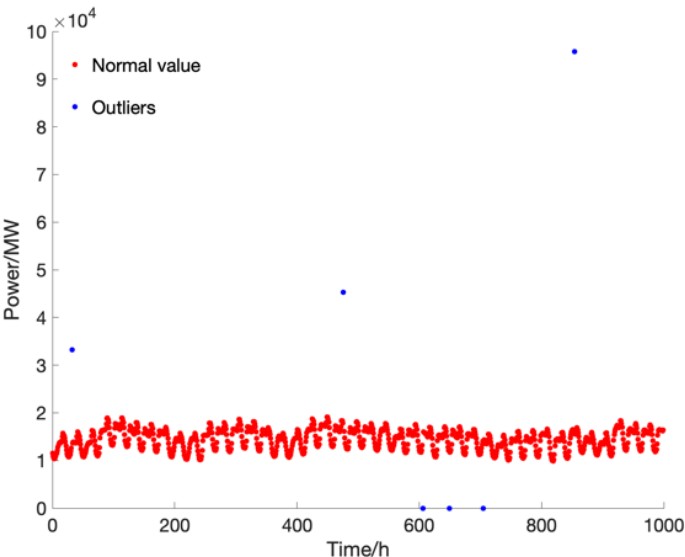

**Figure 6.** DBSCAN clustering diagram.

The root mean square error (RMSE) and mean absolute percentage error (MAPE) are used as the evaluation indicators in this paper. The smaller the value of evaluation indicators, the higher the forecast accuracy.

$$\text{RMSE} = \sqrt{\frac{1}{n}\sum_{i=1}^{n}(y_i - \hat{y}_i)^2} \tag{10}$$

$$\text{MAPE} = \frac{1}{n}\sum_{i=1}^{n}|y_i - \hat{y}_i| \tag{11}$$

where $y_i$ represents the actual data, and $\hat{y}_i$ represents the forecast data.

It is necessary to determine the value of the decomposition quantity to perform VMD decomposition. In general, the value is between 5 and 20 based on the literature. In this experiment, the center frequency value of each component is calculated with the decomposition quantity starting from 5. It can be seen that some important information in the original signal will be filtered when the decomposition quantity is too small, which will result in insufficient precision for the load forecast. However, the center frequencies of adjacent modal components will be too close, adding noise when the decomposition quantity increases.

Table 1 shows the center frequency of each IMF component when the decomposition quantity is between 5 and 10. It can be seen that the center frequency decreases when $K < 8$, which indicates that the IMF mode may be under-decomposed. When $K = 9$, IMF6 and IMF7 have similar modes, so the modal decomposition number $K = 8$ is suitable in the experiment in this paper. The dynamic profile of different component is shown in Figure 7.

**Table 1.** The center frequency of each IMF component under different $K$.

| $K$ | IMF1 | IMF2 | IMF3 | IMF4 | IMF5 | IMF6 | IMF7 | IMF8 | IMF9 | IMF10 |
|---|---|---|---|---|---|---|---|---|---|---|
| 5 | 0.02 | 83.15 | 41.51 | 288.77 | 165.29 | | | | | |
| 6 | 0.02 | 83.14 | 41.51 | 208.46 | 290.39 | 164.28 | | | | |
| 7 | 0.01 | 41.51 | 83.14 | 208.44 | 290.29 | 164.29 | 374.00 | | | |
| 8 | 0.01 | 41.51 | 83.14 | 164.31 | 208.44 | 290.29 | 373.77 | 457.30 | | |
| 9 | 0.01 | 41.53 | 83.11 | 164.32 | 208.40 | 290.19 | 327.33 | 374.47 | 457.41 | |
| 10 | 0.01 | 41.52 | 83.14 | 164.34 | 208.04 | 248.93 | 291.26 | 329.93 | 374.67 | 457.53 |

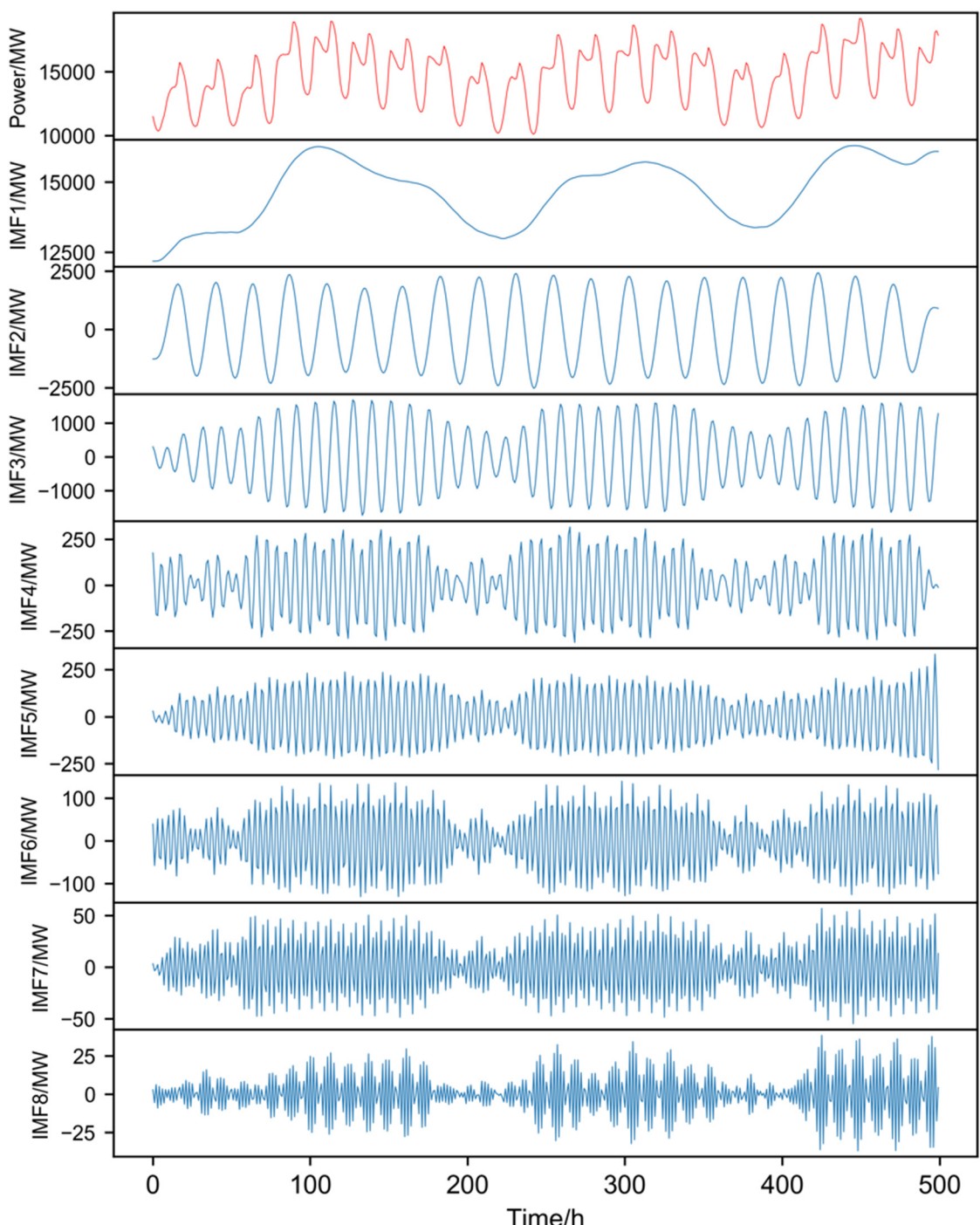

**Figure 7.** Modal component of VMD.

The penalty parameter $\alpha$ is generally set to be about twice the amount of data, and it will be 2000 in this paper. The other parameters will be set as follows: initial center frequency $\omega = 0$, and the convergence parameter is r = $1 \times 10^{-7}$. The first 1000 pieces of data in this data set are decomposed, and the modal components diagram is shown in Figure 6. It can be seen that the modal components decomposed by VMD are smoother than the original data, which is useful and helpful for subsequent neural network training. Among them, the component IMF1 is close to the trend of the original load sequence profile, and it reflects the overall load trend.

### 4.2. Analysis of Complementary Characteristics of Hybrid Algorithms

In order to verify the reliability of the proposed model, the actual operation load data of a certain province are used as data set 1, the data from 0:00 on 1 January 2016 to 18:00 on 26 May 2018 are used as the training data set and a total of 50 h of data from 19:00 on 26 May to 20:00 on 28 May 2018 is used as the test data set; the sampling frequency is 1 h, and the iterative prediction is performed 6 h in advance.

The GEFCom2012 data are used as data set 2. The load data from 0:00 on 1 January 2009 to 23:45 on 25 January 2009 are used as the training data set and the load data at 23:45 on 7 February 2009 are used as the test set; the sampling frequency is 15 min. The iterative prediction is carried out 2 h in advance. The GRU network and TCN network parameters used, respectively, in the experiment are determined by training and testing. The parameters of the two neural networks are shown in Table 2.

**Table 2.** Parameters of neural networks.

| Parameter | GRU | TCN |
|---|---|---|
| hidden layer | 3 | 3 |
| batch size | 128 | 128 |
| epoch | 100 | 100 |
| learning rate | 0.01 | 0.01 |
| dropout | 0.1/0.2 | 0.1/0.2 |
| activation function | relu | relu |
| dilation rate | - | 1/2/4 |

Figure 8 shows the load forecasting results of the eight IMF components in data set 1 using the GRU model and the TCN model, respectively. For the low-frequency load components, the error between actual and forecasting data based on the GRU network is smaller than that of the TCN forecasting model. However, the prediction accuracy of GRU will decrease with the frequency increase such as IMF5 to IMF 8.

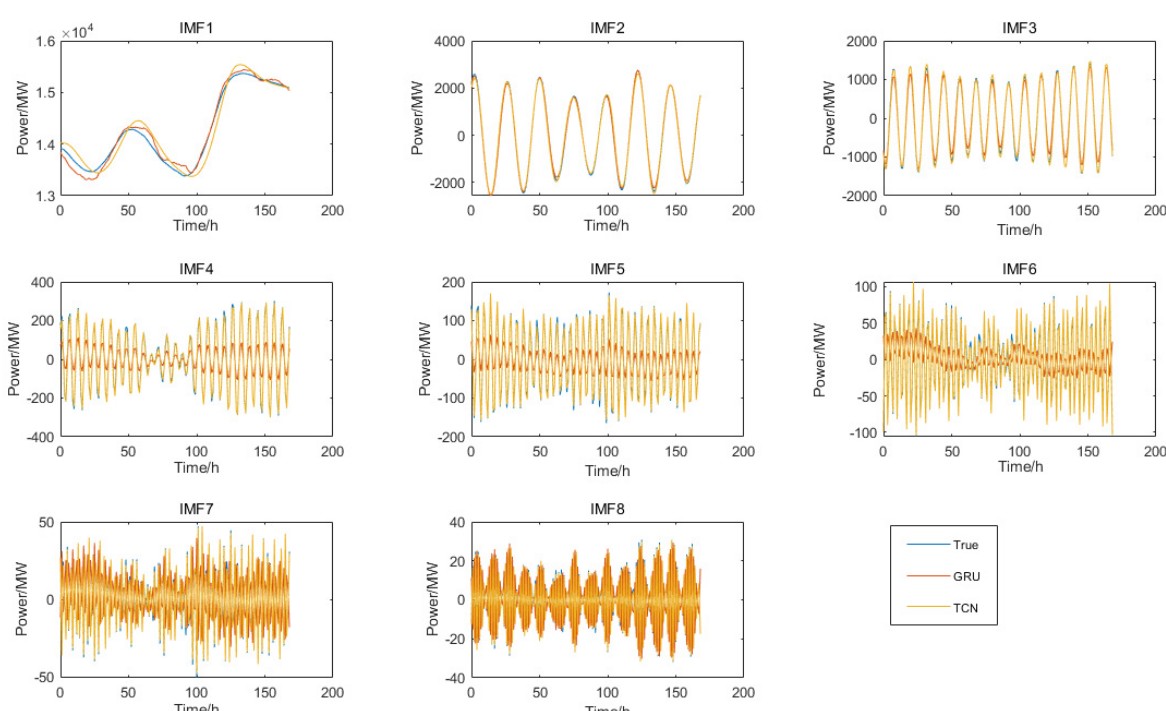

**Figure 8.** The forecasting results of GRU and TCN, respectively, in the high-frequency component.

Furthermore, the GRU model and the TCN model are used for the load forecasting of high-frequency components and low-frequency components in data set 2. The results are shown in Figures 9 and 10, respectively. It can be seen that in the low-frequency component, the forecasting result of the GRU model is more accurate than that of the TCN model. Figure 10 shows that the GRU model has a larger prediction error when the time-series data fluctuate too fast.

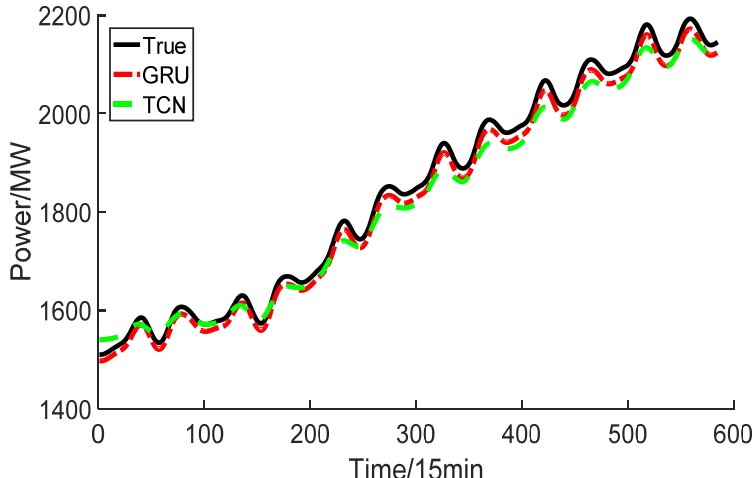

**Figure 9.** The forecasting results of GRU and TCN with low-frequency component.

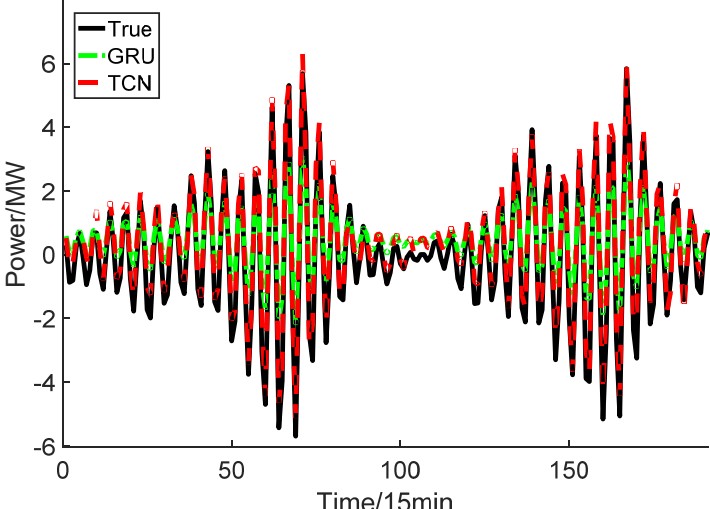

**Figure 10.** The forecasting results of GRU and TCN with high-frequency component.

### 4.3. The Comparison of Different Forecasting Models

In order to verify the validity of the proposed method in this paper, GRU, TCN, LSTM, Prophet, VMD-GRU, and VMD-TCN were selected for experimental comparison, and the forecasting results are shown in Table 3. It can be seen that after the VMD decomposition, the forecasting errors of the GRU and TCN neural networks are lower than that without VMD decomposition. The performance of the forecasting is significantly improved, which indicates the feasibility of using VMD to decompose the original load sequence. Among the individual algorithms, the forecasting error of the GRU model is the smallest with the RMSE of 50.38 MW and MAPE of 1.12% in data set 1.

**Table 3.** Comparison of different forecasting methods.

| Method | Data Set 1 | | | Data Set 2 | | |
|---|---|---|---|---|---|---|
| | RMSE/MW | MAPE | Time/s | RMSE/MW | MAPE | Time/s |
| GRU | 58.43 | 1.19 | 120 | 25.21 | 1.83 | 169 |
| TCN | 77.08 | 1.32 | 131 | 44.24 | 2.50 | 146 |
| LSTM [37] | 64.05 | 1.27 | 157 | 33.54 | 2.09 | 190 |
| Prophet [38] | 144.58 | 4.71 | 172 | 87.25 | 4.41 | 231 |
| XG Boost [38] | 125.23 | 1.68 | 185 | 36.25 | 2.35 | 199 |
| VMD-GRU | 50.38 | 1.12 | 982 | 11.43 | 1.21 | 1292 |
| VMD-TCN | 56.64 | 1.14 | 374 | 24.72 | 1.75 | 495 |
| VMD-GRU-TCN | 32.14 | 0.42 | 731 | 10.19 | 0.97 | 1007 |

Due to the high randomness and non-stationarity of the original load profile, the forecasting model constructed by a separate algorithm cannot fit the actual load situation very well. The results show that the hybrid model combined with the GRU and TCN network is the smallest both in data set 1 and data set 2. The RMSE of the forecasting result of VMD-GRU is reduced by 36.20% and 10.8%, respectively, while the MAPE of the forecasting result of VMD-GRU is reduced by 62.5% and 19.8%.

At the same time, the training and operation time of different prediction models is shown in Table 3. It can be seen that the operation time of the model proposed in this paper is slightly longer than that of the other non-decomposition prediction. The time-series load data are decomposed into eight components by VMD, and the process is serial, which increase the calculation time. However, if the different components are calculated in parallel, then the operation time will be greatly reduced. From Table 3, the TCN network itself has the ability of parallel calculation. Therefore, the computation time of the VMD-GRU-TCN model is reduced compared with the VMD-GRU model. Figures 11 and 12 show the comparison of different models based on data set 1 with 24 sample points and data set 2 with 96 sample points on the next day. It can be seen from Figures 11 and 12 that the load forecasting result has a high similarity with the original load profiles.

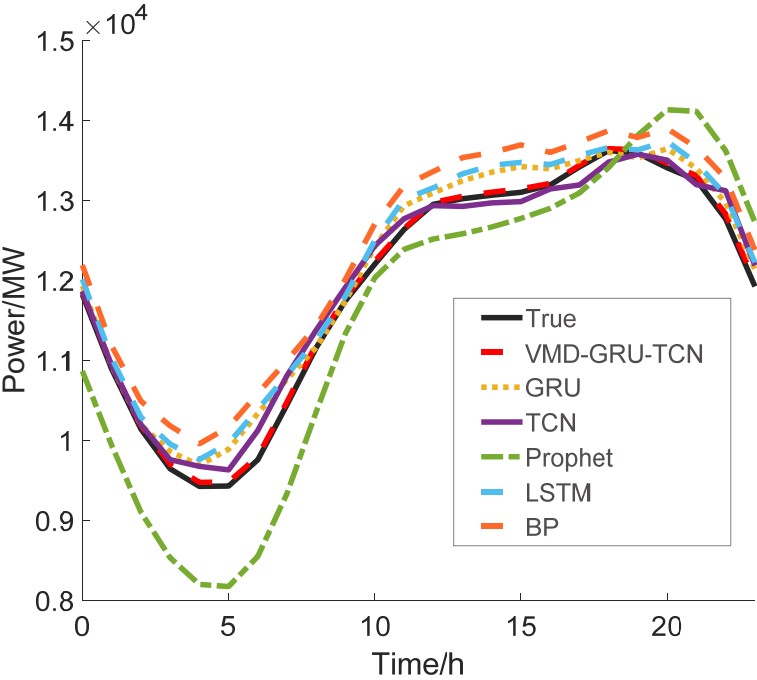

**Figure 11.** The forecasting result of data set 1.

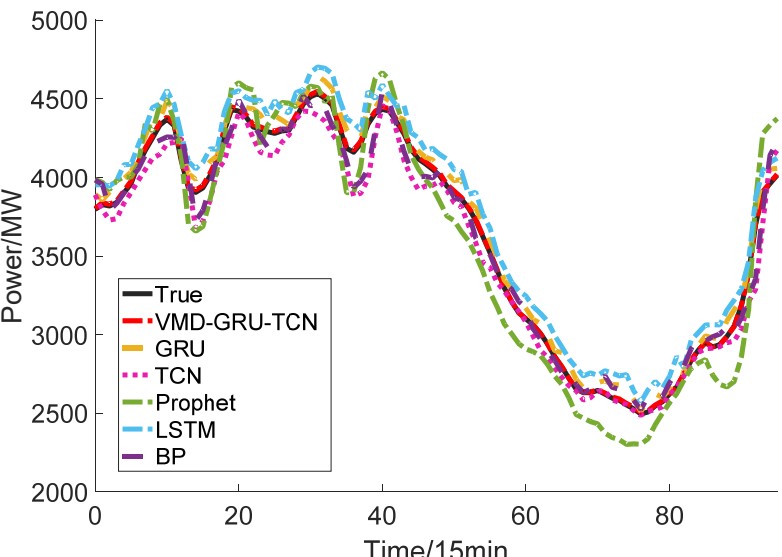

**Figure 12.** The forecasting result of data set 2.

## 5. Conclusions

This paper proposes a short-term load forecasting method based on a VMD and GRU-TCN combined network, which effectively solves the problem of low accuracy of traditional forecasting models due to the high randomness and non-stationarity of original load data. The variational modal decomposition is used to decompose the noisy original load sequence data into intrinsic IMF modal components with different frequencies and amplitudes, which can effectively reduce the complexity of the original time-series data. The gated recurrent unit (GRU) and temporal convolutional network (TCN) are used to extract, train and predict the IMF modal components. The simulation results show that the combined method has a certain degree of improvement in the prediction accuracy compared with the single prediction model.

However, there are some limitations of the proposed method in this paper. (1) The future work will focus on the coordination mechanism of the combined neural network. The forecasting results of different IMF models based on TCN and GRU will be linearly combined together for the result of electrical load forecasting, and the adaptive combined method will be a more suitable strategy. (2) From Table 3, which shows a comparison with the VMD–TCN method, the proposed model needs more time to obtain a better result. Thus, we will build a parallel computing framework for the proposed method in future work.

**Author Contributions:** Data curation, Y.L.; Formal analysis, T.Z.; Resources, Z.S.; Software, Y.H.; Writing—original draft, C.C. All authors have read and agreed to the published version of the manuscript.

**Funding:** This research was funded by "National Natural Science Foundation of China, grant number 51607057", "The Fundamental Research Funds for the Central Universities, grant number 2020B22514"and "The open funding of Jiangsu Key Laboratory of Power Transmission & Distribution Equipment Technology, grant number 2021JSSPD07".

**Institutional Review Board Statement:** Not applicable.

**Informed Consent Statement:** Not applicable.

**Conflicts of Interest:** The authors declare no conflict of interest.

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
