# Peer review of "Short-Term Electrical Load Forecasting Based on VMD and GRU-TCN Hybrid Network"

_applsci, doi:10.3390/app12136647_

Round 1
Reviewer 1 Report
This study suggested a short-term load forecast method that combines Variational Modal Decomposition (VMD), Gated Recurrent Unit (GRU), and Time Convolutional Network (TCN) hybrid network. The simulation results based on actual operation data and simulation data show that the proposed method is reliable and can effectively improve the accuracy of load forecasting. Overall, the manuscript is well written and interesting; however, the following issues should be addressed before the final publication.
1. The citation style is not uniform. Kindly provide the citation in a uniform style.
2. Please briefly describe the results in the abstract in a couple of lines.
3. For Section 1, the authors should provide the comments of the cited papers after introducing each relevant work. What readers require is, by convinced literature review, to understand the clear thinking/consideration of why the proposed approach can reach more convincing results. This is the very contribution from the authors.
4. There are typos and grammatical mistakes in the paper. Proofreading is recommended.
5. Please provide high-quality figures with enlarged labels.
6. At the end of section 1, please provide the structure of the manuscript.
7. The literature review is too short and should be extended. In addition, in the literature review, it is highly recommended to discuss other modeling and forecasting methods to highlight the importance of this topic for the reader of this journal. Some recommendations that can be considered are, (10.3390/en15093423), (10.1109/ACCESS.2021.3100076).
8. Provide details about the data used in the analysis section. It would be better to plot it separately as given in “Short-Term Electricity Demand Forecasting Using Components Estimation Technique”.
9. In section ?, please compare the structure of the Chinese electricity market with European electricity markets. Are they similar or different? For example, compare it with the structure stated in the paper “Forecasting of electricity price through a functional prediction of sale and purchase curves”.
Author Response
Dear Editor:
Thank you very much for your letter and the comments from the referees about our paper submitted to APPLSCI (NO. 1776583)
We have learned much from the reviewers’ comments, which are fair, encouraging and constructive. After carefully studying the comments and your advice, we have made corresponding changes. Our response of the comments is enclosed at the end of this letter. We also add some experiments in this manuscript.
If you have any question about this paper, please don’t hesitate to contact us.
Sincerely yours,
Dr. Changchun Cai
Thanks for your comments on our paper. We have revised the paper according to your comments. We have deleted some repetitive presentation. The main revisions are listed as follows:
Review 1:
This study suggested a short-term load forecast method that combines Variational Modal Decomposition (VMD), Gated Recurrent Unit (GRU), and Time Convolutional Network (TCN) hybrid network. The simulation results based on actual operation data and simulation data show that the proposed method is reliable and can effectively improve the accuracy of load forecasting. Overall, the manuscript is well written and interesting; however, the following issues should be addressed before the final publication.
Point:1:The citation style is not uniform. Kindly provide the citation in a uniform style.
Response 1: Thanks for your advices. Authors have revised the paper carefully and rewritten the citation in the paper. Thank you very much.
Point:2:Please briefly describe the results in the abstract in a couple of lines.
Response 2: Thanks for your advices. Authors have added the results comparison with other method in the abstract. Thank you very much.
Point 3:For Section 1, the authors should provide the comments of the cited papers after introducing each relevant work. What readers require is, by convinced literature review, to understand the clear thinking/consideration of why the proposed approach can reach more convincing results. This is the very contribution from the authors.
Response 3: Thanks for your advices. It is puzzle for the readers that why the authors choose the proposed method in this paper. The authors have given the comments of the cited papers in the paper, and we also added some analysis the experiment to supply enough proof for the proposed method in the paper. we all marked red in the paper, thank you very much.
Point 4:There are typos and grammatical mistakes in the paper. Proofreading is recommended.
Response 4: Thanks for your advices. Authors have revised the paper carefully and fixed the mistakes in the paper. Thank you very much.
Point 5:Please provide high-quality figures with enlarged labels.
Response 5: I am sorry about the figures. Authors have revised each figure and enlarged the labels in the paper especially in figure 8. Thank you very much.
Point 6:At the end of section 1, please provide the structure of the manuscript.
Response 6: Thank you very much, we have added the structure of the manuscript in the end of the section 1. We give the context of each section and the relationship of each section. Thank you very much.
Point 7:The literature review is too short and should be extended. In addition, in the literature review, it is highly recommended to discuss other modeling and forecasting methods to highlight the importance of this topic for the reader of this journal. Some recommendations that can be considered are,(10.3390/en15093423), (10.1109/ACCESS.2021.3100076).
Response 7: Thank you very much, we have added some literature reviews and extend the literature review. Furthermore, some discussion about the modeling and forecasting methods literature is highlight.
Point 8: Provide details about the data used in the analysis section. It would be better to plot it separately as given in “Short-Term Electricity Demand Forecasting Using Components Estimation Technique”.
Response 8: Thank you very much. We have added the detail description about the data in the analysis section 2 about the yearly and daily electrical power load profiles in section 2. Thank you very much.
Point 9: In section ?, please compare the structure of the Chinese electricity market with European electricity markets. Are they similar or different? For example, compare it with the structure stated in the paper “Forecasting of electricity price through a functional prediction of sale and purchase curves”.
Response 9: Thank you very much. The structure of the Chinese electricity market and European electricity markets is very different. We have added some literature and description about the load forecasting consider the electrical price and load purchase curves. Such as “Forecasting of electricity price through a functional prediction of sale and purchase curves”. In this paper, it gives a new approach for prediction of the electricity price based on forecasting aggregated purchase and sale curves. Furthermore, the authors also discuss some about the load forecasting method consider electrical price, purchase curves in the section 1“Introduction”. Thank you very much.

Reviewer 2 Report
Authors proposed a Short-term load forecasting based on VMD and GRU-TCN hybrid network. After reading submitted manuscript, my comments are as follows :
1. It is suggested to include the detail in title, i.e., Short-term load forecasting of what ? Kindly revised the title.
2. VMD is used to decomposed time series signals. How suitable IMF's was selected.It is not appropriate to use all the IMF's for prediction.There are several papers which gives an idea how to select suitable IMF's. Kindly refer following journals and provide justification in revised manuscript :
a. https://journals.sagepub.com/doi/abs/10.1177/09544062211043132.
b. https://www.mdpi.com/2079-6412/12/3/419.
3. It is always a good practice to apply k-fold cross validation procedure for classification, regression as well as forecasting. Kindly address in revised manuscript why only authors focused on training and testing. Refer suitable published journals.
4. Details about experimental data/simulation need to be improved. Kindly give suitable reference also.
5. It is recommended to prepare a comparison table with already published literatures in which various authors utilized the same dataset.It will be useful to highlight the authors contributions.
6. What are the limitations of proposed model. Further future scope should be highlighted.
7. In abstract, kindly include the numerical values of results and include details which algorithms give better results.
8. Recommended including high resolution images in revised manuscript.
Author Response
Dear Editor:
Thank you very much for your letter and the comments from the referees about our paper submitted to APPLSCI (NO. 1776583)
We have learned much from the reviewers’ comments, which are fair, encouraging and constructive. After carefully studying the comments and your advice, we have made corresponding changes. Our response of the comments is enclosed at the end of this letter. We also add some experiments in this manuscript.
If you have any question about this paper, please don’t hesitate to contact us.
Sincerely yours,
Dr. Changchun Cai
Thanks for your comments on our paper. We have revised the paper according to your comments. We have deleted some repetitive presentation. The main revisions are listed as follows:
Review 2:
Authors proposed a Short-term load forecasting based on VMD and GRU-TCN hybrid network. After reading submitted manuscript, my comments are as follows :
Point 1:It is suggested to include the detail in title, i.e., Short-term load forecasting of what ? Kindly revised the title.
Response 1: Thank you very much. You are right, it will be better with the title “Short-term electrical load forecasting based on VMD and GRU-TCN hybrid network”. Thank you.
Point 2:VMD is used to decomposed time series signals. How suitable IMF's was selected. It is not appropriate to use all the IMF's for prediction. There are several papers which gives an idea how to select suitable IMF's. Kindly refer following journals and provide justification in revised manuscript :
- https://journals.sagepub.com/doi/abs/10.1177/09544062211043132.
- https://www.mdpi.com/2079-6412/12/3/419.
Response 2: Thank you very much. In this paper we choose the suitable IMF components based on the conclusion of the IMF decomposition. Based on the discussion of the center frequency of each IMF component with the different decomposition number in section 4.1. The separated degree is the best when the number is 8. We also added some refs about the VMD method. Thank you very much.
Point 3: It is always a good practice to apply k-fold cross validation procedure for classification, regression as well as forecasting. Kindly address in revised manuscript why only authors focused on training and testing. Refer suitable published journals.
Response 3: Thank you very much. As you mentioned, k-fold cross validation procedure for classification, regression in widely applied in forecasting and have a better performance. Authors have added suitable published journals in the ref in section 1
Point 4: Details about experimental data/simulation need to be improved. Kindly give suitable reference also.
Response 4: Thank you very much. Authors have added the references about the experimental data. There are two datasets in this paper, GEFCom2012 is hierarchical load forecasting and wind power forecasting, a summary of the methods used by selected top entries. Another dataset is an operational data of actual power grid in guizhou province in china. We added description about the dataset in the paper. thank you very much.
Point 5: It is recommended to prepare a comparison table with already published literatures in which various authors utilized the same dataset. It will be useful to highlight the authors contributions.
Response 5: Thank you very much. A comparison table with already published literatures in which various authors utilized the same dataset1 is shown Table III in the paper. Thank you very much.
Point 6: What are the limitations of proposed model. Further future scope should be highlighted.
Response 6: Thank you very much. Yes, there are some limitations of the proposed model in the paper, and we want to overcome the shortage in the future work. Authors have added future work and shortage in the conclusion. “However, there are some limitations about the proposed method in this paper. (1) The future work will force on the coordination mechanism of the combined neural network. The forecasting results of different IMF based on TCN and GRU will linear combine together for the result of electrical load forecasting, and the adaptive combine method will be more suitable strategy. (2) From Table III, comparison with VMD-TCN method, the proposed need more time to get a better result, we will build a parallel computing framework for the proposed method in the future work.”
Point 7: In abstract, kindly include the numerical values of results and include details which algorithms give better results.
Response 7: Thank you very much. The authors have added some discussion about the results in the abstract and given the comparison about different algorithm mentioned in the paper.
Point 8: Recommended including high resolution images in revised manuscript.
Response 8: Thank you very much. The authors have read all the images and redraw the images to improve the readability. Thank you.

Round 2
Reviewer 1 Report
The authors have addressed my concerns and hence, I recommend the paper for publication in its present form.
Reviewer 2 Report
Authors have addressed all reviewer comments with justifications and accordingly modified the manuscript.